# An observational study of intensivists' expectations and effects of fluid boluses in critically ill patients

Olof Wall[1,2]*, Salvatore Cutuli[3,4], Anthony Wilson[4,5], Glenn Eastwood[4], Adam Lipka-Falck[6], Daniel Törnberg[2,7], Rinaldo Bellomo[4], Maria Cronhjort[1,6]

1 Department of Clinical Science and Education, Karolinska Institutet, Södersjukhuset, Stockholm, Sweden, 2 Department of Anaesthesiology and Intensive Care, Danderyds Sjukhus, Stockholm, Sweden, 3 Dipartimento di Scienze dell' Emergenza, Anestesiologiche e della Rianimazione, Fondazione Policlinico Universitario Agostino Gemelli IRCCS, Rome, Italy, 4 Department of Intensive Care, Austin Hospital, Melbourne, VIC, Australia, 5 Adult Critical Care, Manchester University Hospitals NHS Foundation Trust, Manchester, United Kingdom, 6 Department of Anaesthesiology and Intensive Care, Södersjukhuset, Stockholm, Sweden, 7 Department of Clinical Sciences, Karolinska Institutet, Danderyd Hospital, Stockholm, Sweden

* olof.wall@ki.se

**Data Availability Statement:** The datasets from this study cannot be shared openly due to confidentiality, per the ethical approval obtained. Requests for access to the data should be made to

## Abstract

### Background

Fluid bolus therapy (FBT) is common in ICUs but whether it achieves the effects expected by intensivists remains uncertain. We aimed to describe intensivists' expectations and compare them to the actual physiological effects.

### Methods

We evaluated 77 patients in two ICUs (Sweden and Australia). We included patients prescribed a FBT ≥250 ml over ≤30 minutes. The intensivist completed a questionnaire on triggers for and expected responses to FBT. We compared expected with actual values at FBT completion and after one hour.

### Results

Median bolus size (IQR) was 300 ml (250–500) given over a median (IQR) of 21 minutes (15–30 mins). Boluses were 57% Ringer´s Acetate and 43% albumin (40-50g/L). Hypotension was the most common trigger (47%), followed by oliguria (21%). During FBT, 55% of patients received noradrenaline and 38% propofol. Intensivists expected a median MAP increase of 2.6 mmHg (IQR: -3.1 to +6.8) at end of bolus and of 1.3 mmHg (-3.5 to + 4.1) after one hour. Intensivist´s' expectations were judged to be accurate if they were within 5% above or below measured values. At FBT completion, 33% of MAP expectations were overestimations and 42% were underestimations. One hour later, 19% were overestimations and 43% were underestimations. Only 8% of expectations of measured urine output (UO) were accurate and 44% were overestimations. Correction for sedation or vasopressors did not modify these findings.

the Research Data Office at Karolinska Institutet via rdo@ki.se and if permitted by law and ethical approval, decided on a case by case basis, the data can shared for researchers who meet the criteria for access.

**Funding:** OW was supported for the study by the Stockholm County Council combined clinical residency and PhD training program. https://www.sll.se/om-regionstockholm/forskning-och-innovation/forskning-och-utveckling/ The funders had no role in study design, data collection and analysis, decision to publish, or preparation of the manuscript.

**Competing interests:** The authors have declared that no competing interests exist.

**Abbreviations:** FB, Fluid bolus; FBT, Fluid bolus therapy; MAP, Mean arterial pressure; SBP, Systolic blood pressure; DBP, Diastolic blood pressure; HR, Heart rate; CVP, Central venous pressure; ScvO2, Central venous oxygen saturation; UO, Urine output; CI, Cardiac Index; SvO2, Mixed venous oxygen saturation; NA, Noradrenaline; ICU, Intensive care unit.

## Conclusions

The physiological expectations of intensivists after FBT carried a high risk of both over and underestimation. Since the physiological effect FBT was often small and did not meet clinical expectations, a reassessment of its rationale, effect, duration, and role appears justified.

## Introduction

Fluid bolus therapy (FBT) is common in critically ill patients [1, 2]. Dynamic indices of preload (e.g., stroke volume variation, pulse pressure variation, vena cava variability) appear to have acceptable predictive values for the immediate post bolus response but, due to practical limitations, they are often not available. Instead, intensivists use a broad range of measures, such as hypotension, tachycardia, oliguria or lactate levels, to inform the decision to administer FBT [3–5]. The FENICE trial described indications for FBT in critically ill patients, with 59% of FBT initiated because of hypotension, followed by oliguria and lactate clearance. The only commonly used hemodynamic measure was central venous pressure (CVP) (25%), which has been shown to be low value in guiding fluid therapy [6]. There is also no clear consensus on which patients will respond to FBT or on the optimal rate of volume and infusion rate of a fluid bolus (FB) [3, 7–10]. Finally, how the decision-making is performed and what response ICU-practitioners expect from a FB has not been investigated [3, 8–10]. As a positive fluid balance is associated with increased mortality in intensive care unit (ICU) patients [7, 11], it appears desirable to study current practice patterns for FBT and what factor influence them so that unnecessary FBT could be avoided. The relationship between intensivists' expectations and actual quantitative hemodynamic effects may help guide this process but has never been described.

As a new way of describing the practice and rationale for FBT, we aimed to compare the quantitative expectations of treating intensivists with the actual effects of FBT in critically ill patients. We aimed to do this in a pragmatic fashion in which treatment modalities and goals for FBT were left at the intensivist discretion.

## Material and methods

### Study design

This was a prospective, observational multi-center cohort study conducted in a Swedish tertiary center (Södersjukhuset) and an Australian university teaching hospital (Austin Hospital). Patients were included between May and September 2017 in Australia and in Sweden between October 2017 to October 2018 and March to April 2019 due to availability of technical equipment to extract data only during these timeframes limiting possible inclusions. The planned study population was 100 patients, but inclusion stopped at 77 since the equipment was not available to use for further study inclusions.

Ethical approval was obtained from the Ethical Review Board of Stockholm (Tomtebodavägen 18A Solna, Sweden), EPN 2017/1133-31 on 4/8/2017 and the Austin Health Human Research Ethics Committee (145 Studley Road Heidelberg Victoria, Australia), LNR/17/Austin/94 on 17/5/2017. The study was registered on Clinicaltrials.gov, NCT03178578. The need for informed consent was waived with permission from the Ethical Review Board. Consent for publication was not applicable.

Inclusion criteria: Admission to the ICU and age 18 years or older. The clinician has pre-scribed a FB of $\geq$250 ml in $\leq$30 minutes. Exclusion criteria: Patients in whom death was considered imminent (within 24 hours) and/or if the treating intensivist declined to participate.

## Treatment

When the treating intensivist chose to give a FB (defined as 250 mL or more of fluids over 30 minutes or less, according to intensivist's choice), the patient was included. The decision to initiate FBT, the volume, rate of infusion and type of fluid was left entirely to the discretion of the treating physician. No specific monitoring or test of responsiveness was mandated but was left to the intensivist's discretion, as this was part of the study outcome. The intensivist then completed a questionnaire (see S1 Appendix) and selected a primary and a secondary FBT trigger from a pre-specified list: hypotension, tachycardia, oliguria, low CVP, high lactate levels, low mixed venous oxygen saturation ($SvO_2$) or central venous oxygen saturation ($ScVO_2$) and low cardiac index (CI).

The questionnaire also asked the intensivist to specify what changes in these variables they expected at completion of the bolus and one hour later. Only the expectations and actual results of the first bolus (at any time in ICU-course) after inclusion were measured and any further boluses given to the same patient were not considered. Patients received all care at the discretion of the treating intensivist, including all measurements and monitoring.

## Monitoring

Data was extracted from the medical information systems Clinisoft® (GE Healthcare, Bar-ringgton, Illinois, USA, TakeCare® (CompuGroup Medical, Koblenz, Germany) or Power-Chart® (Cerner Corporation, Kansas City, Missouri, USA) regarding mean arterial pressure (MAP), heart rate (HR), CI, CVP, urine output per hour (UO) and laboratory values such as lactate, $SvO_2$ or $ScVO_2$ and creatinine for comparison. Monitoring equipment was IntelliVue MP70® and IntelliVue MX800® (Philips Healthcare, Eindhoven, Netherlands). Monitoring and laboratory values were ordered at the intensivist's discretion and per departmental protocol, meaning that there was no mandated measurement of any advanced hemodynamic values, and these would be used as clinically indicated. Departmental standards for blood pressure measurement included intraarterial measure via arterial line (Merit Medical, South Jordan, UT, USA and ITL Biomedical, Mulgrave, Australia) and for cardiac output measures include calibrated thermodilution PiCCO® (Pulsion Medical Systems, Feldkirchen, Germany) and pulmonary artery catheter (Edwards Lifesciences, Irvine, CA, USA). Measurements were collected for 5 minutes before the onset of the fluid bolus as a baseline and until 60 min after completion of the fluid bolus. Any infusions of vasoactive medications, sedatives and diuretics given were also recorded and accounted for as confounders for hemodynamic changes. Any further fluids administered during the study period were registered and accounted for as confounders. Fluids were recorded as colloids, crystalloids, or maintenance fluids.

## Outcomes

**Primary outcome.** Accuracy of the intensivist's expectations on the physiological effect of FBT at completion of the FB.

**Secondary outcomes.** To describe the accuracy of the intensivist's expectations of the physiological effects of FBT one hour after the completion of the FB. The trigger for the FBT and changes in CO, HR, MAP, CVP, and lactate, UO, $ScvO_2$ or SvO2.

## Statistical analysis

We studied a convenience sample, planned to consist of 100 patients but resulting in 77 patients. Normality was tested using the Shapiro-Wilk test. Categorical data was presented as fractions and percentages and continuous data was presented as means with SD or median with IQR, and 95% CI depending on underlying distribution. Linear regression models were performed to determine relationship between the levels of change in sedation and vasoactive medications and on each of the variables used by the clinicians, and a correction was then made for the relevant levels of sedation and vasoactive medications. Imputation for missing data was not performed as the datasets were largely complete.

Intensivist´s' expectations were judged to be accurate if they were within 5% above or below measured values, as the authors judged this to be a clinically relevant interval. For purposes of defining clinical effectiveness and whether expectations were under- or overestimations, MAP, CO, CVP, $ScvO_2$ and UO were expected to increase after FBT, and HR and lactate to decrease.

Bland-Altman plots were used to describe the relationship between measured and expected values. We performed unplanned subgroup analyses on the patients where the indication for a FB was hypotension, to explore reasons for the unexpectedly low expectations in this group. An alpha value of 0.05 was considered statistically significant.

Statistical analysis was performed using IBM SPSS Statistics® version 25 for Windows (IBM Co., Armonk, NY, USA.

## Results

We studied 77 patients, 18 in Australia and 59 in Sweden (see **Fig 1** for study flow chart). Of these, 46 were male, with a median age of 68 years and a median SOFA score of 8 (**Table 1**). 24 patients were monitored with invasive hemodynamic monitoring (other than invasive blood pressure monitoring), and 44 patients were receiving mechanical ventilation. Bolus fluid was either Ringers Acetate (57%) and albumin 40-50g/L (43%), with a median (IQR) volume of 300ml (250 to 500) ml and a median (IQR) duration of administration time of 22 minutes (15 to 30 mins). 42% of participating intensivists were ICU consultants, 15% were fellows and 43% were registrars.

Expectations of MAP were accurate in 25% of cases at FBT completion and in 37% of cases after one hour. At FBT completion, 33% of MAP expectations were overestimations and 42% were underestimations. A scatter plot of the relationship between expected and measured MAP at fluid bolus completion by main reason for bolus is illustrated in **Fig 2**. One hour later, 19% were overestimation and 43% were underestimations. For HR, expectations were accurate in 52% of cases at the end of the bolus and 39% of cases after one hour. In 38% of cases, expectations overestimated the effect at FBT completion, and in 39% after one hour. In the case of UO, expectations were accurate in only 8% of cases after one hour, 44% were overestimation and 48% were underestimations (**Table 2**). Scatter plots of the relationship between expected and measured UO by main reason for bolus at fluid bolus completion is illustrated in **Fig 3**.

Intensivists expected a median MAP increase of 2.6 mmHg (IQR: -3.1 to +6.8 mmHg) at end of the FB and of 1.3 mmHg (-3.5 to + 4.1 mmHg) after one hour. Expectations for CI were a median (IQR) increase of 0.06 L/min/m$^2$ (-0.05 to +0. L/min/m$^2$) at bolus completion and a median increase of 0.00 L/min/m$^2$ (-0.12 to +0.20 L/min/m$^2$) after one hour. Expectations for HR were a median (IQR) decrease of -4.2 bpm (-11.0 to 0.0 bpm) at bolus completion and a median decrease of -3.7 bpm (-9.6 to +0.2 bpm) after one hour. Expectations are further described in **Table 3**. Expectations for the primary reason for FBT are described in **S1 Table** in

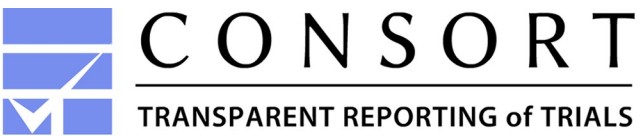

## CONSORT 2010 Flow Diagram

**Enrollment**

Assessed for eligibility when
prescribed a fluid bolus n=94

Excluded (n=17)
- Incomplete questionnaire (n=2)
- Actual bolus given not corresponding
  to criteria (n= 7)
- Technical issues with data retrieval
  from monitors (n= 4)
- Left ICU during monitoring (n=1)
- Other reasons (n=3)

Included n=77

**Allocation**

Allocated to intervention n= 77
- Received allocated intervention n= 77
- Did not receive allocated intervention  n=0

**Follow-Up**

Lost to follow-up n= 0

Discontinued intervention n=0

**Analysis**

Analysed n= 77
- Excluded from analysis n= 0

**Fig 1. Study flow chart.** Consort flow chart of patients screened.

**Table 1. Patient characteristics.**

| | |
|---|---|
| Age (years) | 68 (58–78) |
| Male | 46/77 (60%) |
| Height (cm) | 171 (166–178) |
| Weight (kg) | 76 (63–88) |
| SOFA | 8 (6–10) |
| Comorbidities | |
| *Atrial fibrillation* | 9/77 (12%) |
| *COPD* | 14/77 (18%) |
| *Chronic kidney disease* | 6/77 (8%) |
| *Diabetes* | 14/77 (18%) |
| *Hypertension* | 34/77 (44%) |
| *Ischemic heart disease* | 23/77 (30%) |
| *Congestive heart failure* | 5/77 (6%) |
| *Smoking* | 15/77 (20%) |
| Acute admission | 57/77 (74%) |
| Surgical admission | 57/77 (74%) |
| Type of surgery | |
| *Thoracic* | 19/77 (25%) |
| *Abdominal* | 18/77 (23%) |
| *Orthopedic* | 4/77 (5%) |
| *Vascular* | 3/77 (4%) |
| *Other* | 3/77 (4%) |
| *Surgical admission that did not undergo surgery* | 30/77 (39%) |
| Noradrenaline infusion | 42/77 (54%) |
| Noradrenaline dose baseline (Δg/kg/min) | 0.08 (0.03–0.15) |
| Propofol infusion | 29/77 (38%) |
| Propofol dose baseline (mg/kg/h) | 1.33 (1.00–1.84) |

Values are presented as median with (IQR) or numbers (percentages) of patients.

SOFA = sequential organ failure assessment score. COPD = Chronic obstructive pulmonary disease.

S1 Appendix. Boxplots of the relationship between expected and measured MAP as well as UO by main indication for bolus are displayed in **Figs 4 and 5**.

Among patients where the indication for FBT was hypotension, 58% were on a noradrenaline (NA) infusion, at bolus completion, however, the MAP expectation for patients without NA was 1.5 (-3.0 to +6.2) vs. 3.3 (-3.9 to +10.0) with NA. In detail, for 5/36 of these patients, the intensivist had very small expectations of MAP increment (expected change between -2 and -2mmHg). The baseline MAP for these five patients was 69.0 (60.3 to 74.5) mmHg. Two of these had infusions of NA, and levels of NA was unchanged after FBT.

The most common primary trigger for FBT was hypotension (47%), followed by low urine output (21%). The most common secondary triggers were low urine output and tachycardia (both 17%) (**Table 3**).

For the primary reason for bolus administration, the estimation was accurate in 22% of cases at FBT completion and 47% were overestimations. After one hour, the effect estimation for the primary reason for bolus administration was accurate in 29% of cases and 31% were overestimations. For the secondary reason, accuracy was 20% at FBT completion, with 43% being overestimations. After one hour the estimation for the secondary reason was accurate in 22% of cases, with 31% being overestimations.

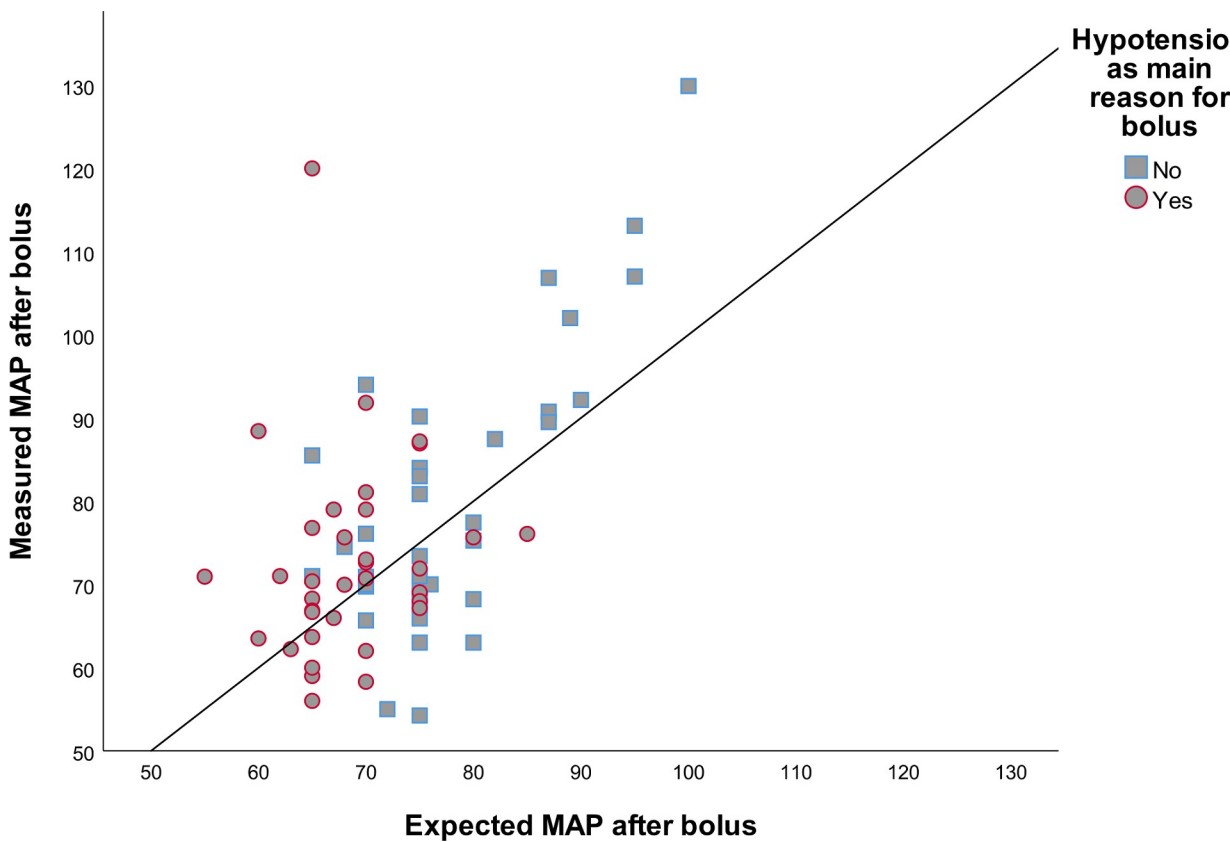

**Fig 2. Scatter plot of measured and expected MAP after fluid bolus.** Scatter plot comparing measured and expected MAP after fluid bolus. Line represents perfect fit. MAP = Mean arterial pressure.

Scatter plots for MAP one hour after FB and HR at both time points are presented in **S1–S3 Figs** in S1 Appendix. All show limited correlation. Bland-Altman plots of bias and limits of agreement, between measured and expected MAP and HR after FB and after one hour as well as UO after one hour are displayed in **S4–S8 Figs** in S1 Appendix.

**Table 2. Accuracy of expectations.**

| Parameter | After bolus | | | One hour after bolus | | |
|---|---|---|---|---|---|---|
| | Correct estimates (%) | Overestimation of effect | Underestimation of effect | Correct estimates (%) | Overestimation of effect | Underestimation of effect |
| MAP (mmHg) | 18/72 (25%) | 24/72 (33%) | 30/72 (42%) | 27/72 (38%) | 14/72 (19%) | 31/72 (43%) |
| UO (ml) | N/A | N/A | N/A | 5/61 (8%) | 27/61 (44%) | 29/61 (48%) |
| HR (bpm) | 38/73 (52%) | 28/73 (38%) | 7/73 (10%) | 29/74 (39%) | 29/74 (39%) | 16/74 (22%) |
| CI (L/min/m²) | 6/23 (26%) | 12/23 (52%) | 5/23 (22%) | 4/22 (18%) | 6/22 (27%) | 12/22 (54%) |
| Lactate (mmol/L) | 2/22 (9%) | 14/22 (64%) | 6/22 (27%) | 2/33 (6%) | 18/33 (54%) | 13/33 (39%) |
| CVP (mmHg) | 5/21 (24%) | 3/21 (14%) | 13/21 (62%) | 3/21 (14%) | 8/21 (38%) | 10/21 (48%) |
| ScvO₂ (%) | 2/3 (67%) | 0/3 (0%) | 1/3 (33%) | 1/3 (33%) | 2/3 (67%) | 0/3 (0%) |

Values are presented as numbers (percentages) of patients.

MAP = Mean arterial pressure. UO = Urine output. HR = Heart rate. CI = Cardiac index. CVP = Central venous pressure. ScvO2 = Central venous oxygen saturation. Overestimation is defined as an estimate greater than measured value for MAP, UO, CI, CVP and ScvO2, and as a measured value greater than estimated value for HR and lactate.

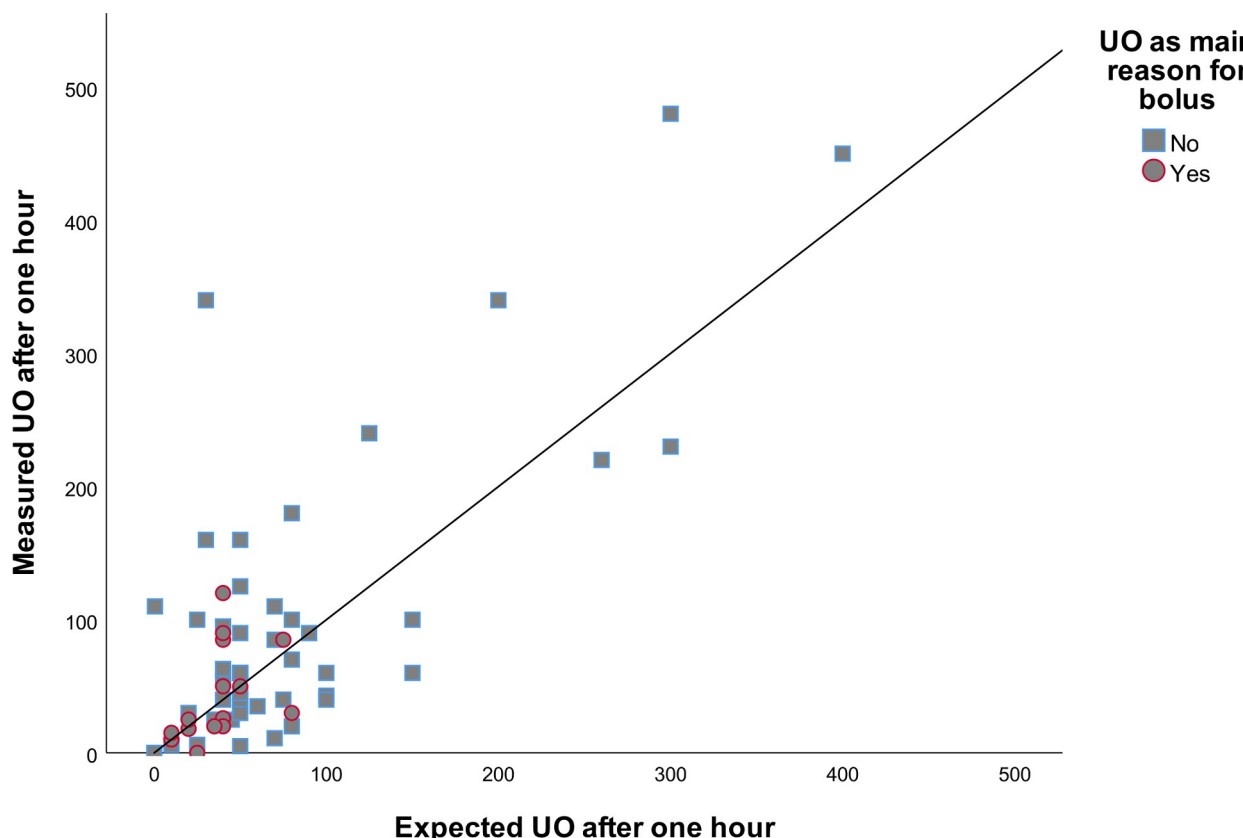

**Fig 3. Scatter plot of measured and expected UO one hour after fluid bolus.** Scatter plot comparing measured and expected UO one hour after fluid bolus. Line represents perfect fit. UO = Urine output.

The median (IQR) MAP change after a fluid bolus was 4.4 (-2.3 to +9.7) mmHg and, after one hour, 2.5 (0.7 to 6.6) mmHg. The median (IQR) HR response after a fluid bolus was -1.0 (-5.4 to +1.5) bpm and after one hour -0.2 (-5.2 to +2.9) bpm. The median (IQR) UO response to fluids after one hour was -15.0 (-60.0 to +15.0) ml (see **Table 4**).

**Table 3. Triggers for FBT and physiological expectations.**

| Reasons for bolus | Main reason, N (%) | Secondary reason, N (%) | Expectation, median (IQR) | |
|---|---|---|---|---|
| | | | After bolus | One hour after bolus |
| Hypotension (mmHg) | 36/77 (47%) | 12/77 (16%) | 2.6 (-3.1–6.8) | 1.3 (-3.5–4.1) |
| Poor urine output (ml) | 16/77 (21%) | 13/77 (17%) | N/A | 0.0 (-47.5–20.0) |
| Tachycardia (bpm) | 14/77 (18%) | 13/77 (17%) | -4.2 (-11.0–0.0) | -3.7 (-9.6–0.2) |
| Low cardiac index (L/min/m$^2$) | 5/77 (6%) | 5/77 (6%) | 0.06 (-0.05–0.31) | 0.00 (-0.12–0.20) |
| High lactate (mmol/L) | 4/77 (5%) | 6/77 (8%) | 0.0 (-0.2–0.3) | -0.2 (-0.4–0.2) |
| Low CVP (mmHg) | 2/77 (3%) | 3/77 (4%) | 0.5 (-0.7–2.4) | 0.0 (-1.1–1.6) |
| Low SvO$_2$/ ScvO$_2$ (%) | 0/77 (0%) | 1/77 (1%) | Too small sample size | Too small sample size |
| None given | N/A | 24/77 (31%) | N/A | N/A |

Values are presented as median with (IQR) or numbers (percentages) of patients.

MAP = Mean arterial pressure. UO = Urine output. HR = Heart rate. CI = Cardiac index. CVP = Central venous pressure. ScvO2 = Central venous oxygen saturation. N/A = Not applicable.

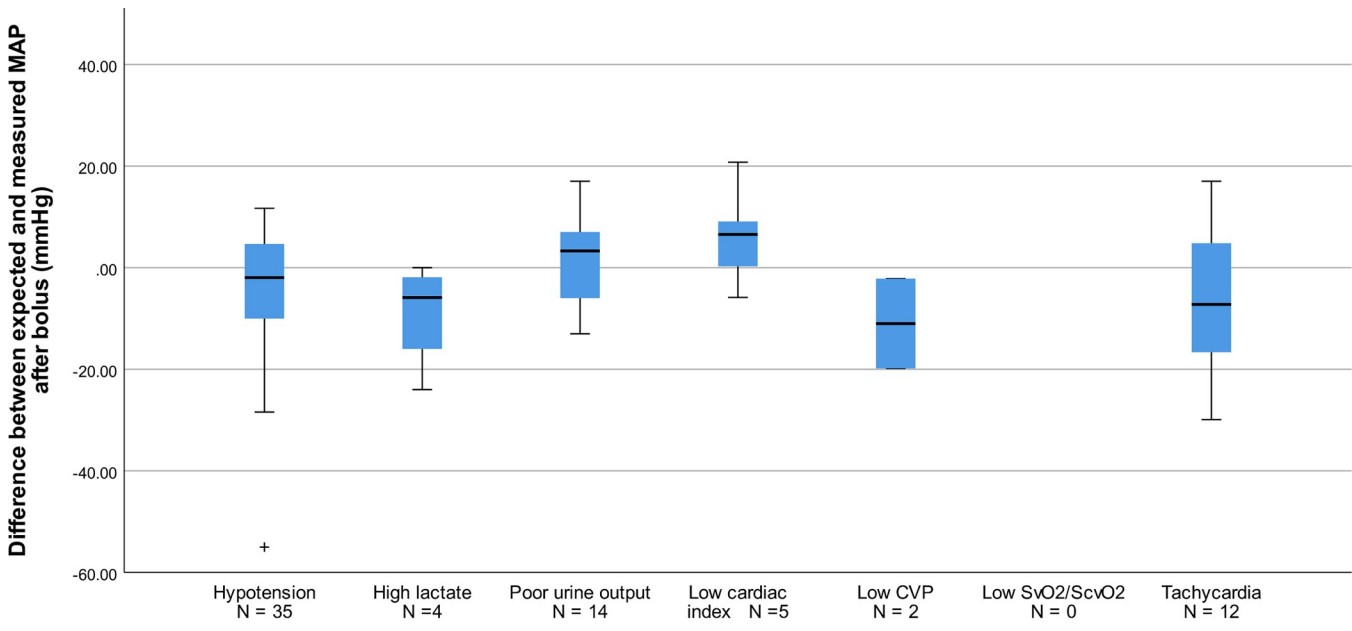

**Fig 4. Boxplot of difference between measured and expected MAP by indication for bolus.** Boxplot showing difference between expected and measured MAP by main reason for administration of fluid bolus. Positive values indicate a higher value for expected MAP compared to measured, and the inverse for negative values. MAP = Mean arterial pressure. CVP = Central venous pressure. ScvO2 = Central venous oxygen saturation. ScvO2 = Mixed venous saturation.

Median (IQR) volume of other fluids was 70 (16 to 113) ml. In addition, 3% of patients were on CRRT and 5% of patients received diuretics within one hour after the fluid bolus. Two patients received boluses of 40 mg and 10 mg Furosemide respectively, one patient had an infusion of 17mg/h continuously during the period and one patient had an infusion of 20 mg/h started. Median (IQR) change in NA was 0.00 (0.00 to 0.00) after the bolus and 0.00 (-0.01 to +0.01) one hour after the bolus, which could otherwise have been a confounder for evaluating response regarding MAP. Six patients were treated with an inotrope during the study period (Four had infusions of milrinone, one dobutamine and one adrenaline), without any dose changes.

The median difference between expectations and outcome is presented in **S2 Table** in S1 Appendix. All values after correction for levels of NA and propofol are presented in **S3 Table** in S1 Appendix as they all were either not statistically significant or did not differ from the uncorrected values. ScvO2 or SvO2 was only measured in 3 patients and the sample size was therefore too small to make adequate analyses.

## Discussion

### Key findings

In this study, the most common triggers for FBT were hypotension, oliguria, and tachycardia. However, in only 25% of cases were expectations of MAP met, underestimation of effect occurred for 42% of boluses, and overestimation for 33%. The clinical expectations of intensivists in relation to urinary output were not met in >90% of cases. These results did not change significantly after correction for administered sedation and vasopressors. Finally, the predictive accuracy of the intensivists' expectations both directly after FBT and at the follow-up one hour later was low. Regardless of accuracy, stated expectations were generally surprisingly small, yet FBT therapy was still initiated.

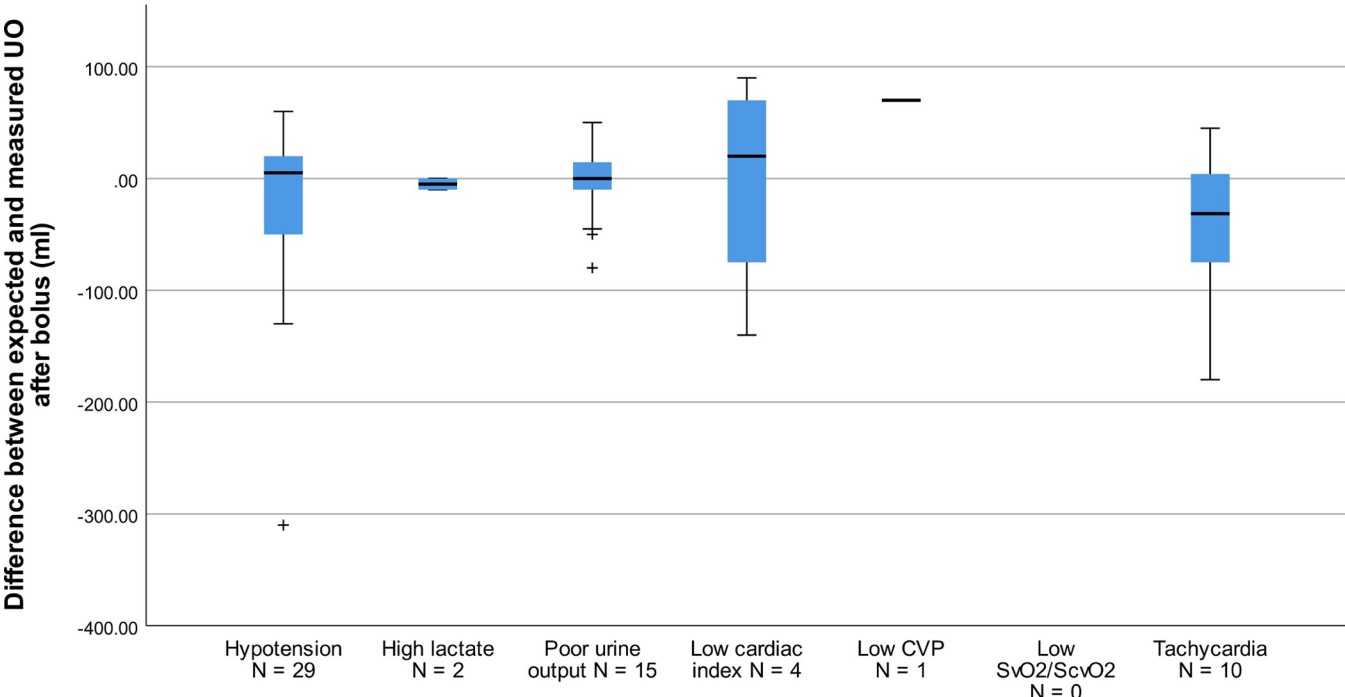

**Fig 5. Boxplot of difference between measured and expected UO by indication for bolus.** Boxplot showing difference between expected and measured MAP by main reason for administration of fluid bolus. Positive values indicate a higher value for expected UO compared to measured, and the inverse for negative values. UO = Urine output. CVP = Central venous pressure. ScvO2 = Central venous oxygen saturation. ScvO2 = Mixed venous saturation.

## Study implications

To our knowledge this is the first study to compare intensivists' quantitative expectations with the actual response to a fluid bolus (FB). Our findings imply that intensivists' expectations of FBT effects are often inaccurate for MAP and markedly so for UO and that, for these two variables, both overestimation and underestimation of effect are common. Intensivists' expectations generally correlate poorly or very poorly with the actual effect. This emphasizes the difficulty in predicting response to fluids, and that many FB interventions will be given without achieving the intended goal. Finally, our study suggests that, as in other studies, the most common triggers for FBT in the ICU remain hypotension, oliguria, and tachycardia.

Expectations were surprisingly small, and could even be negative for several variables, including MAP, the most common indication for a FB. There might be several explanations for this. If the patient is unstable, merely avoiding a further decrease might seem like a reasonable expectation. For a hypotensive patient on vasopressors, expectation of a MAP-response to a FB might manifest as a reduced dose of NA instead of a higher MAP, as this will be titrated down. However, vasopressor doses were not in fact decreased, and the expectations for MAP among patients receiving NA were higher rather than lower.

Observational bias might also be a factor, as knowledge of being monitored might cause intensivists to lower their stated expectations, for fear that they might be found markedly inaccurate [12]. Another explanation might be that FBT is a norm in the ICU environment and intensivists feel there is peer pressure to use this therapy even in cases where their clinical judgment is that the response will be small. Another reason that intensivists feel comfortable with small expectations might be that, at the bedside, a FB might seem like a harmless intervention,

**Table 4. Effects of the fluid bolus.**

| Parameter | Baseline | Difference from baseline after bolus, median (IQR) | Difference from baseline after 1 hour, median (IQR) |
|---|---|---|---|
| MAP (mmHg) N baseline = 77 N after bolus = 75 N after 1 hour = 75 | 69.7 (64.2–78.0) | 4.4 (-2.3–9.7) | 2.5 (-2.2–10.0) |
| UO (ml/h) N baseline = 73 N after 1 hour = 71 | 60.0 (25.0–150.0) | N/A | -15.0 (-60.0–15.0) |
| HR (bpm) N baseline = 77 N after bolus = 76 N after 1 hour = 77 | 92.9 (79.0–111.0) | -1.0 (-5.4–1.5) | -0.2 (-5.2–2.9) |
| CI (L/min/m$^2$) N baseline = 24 N after bolus = 23 N after 1 hour = 22 | 2.68 (2.14–3.10) | 0.00 (-0.13–0.09) | 0.17 (-0.12–0.37) |
| Lactate (mmol/L) N baseline = 68 N after bolus = 28 N after 1 hour = 48 | 1.3 (0.9–1.9) | 0.0 (-0.3–0.1) | -0.1 (-0.3–0.2) |
| CVP (mmHg) N baseline = 23 N after bolus = 21 N after 1 hour = 23 | 10.3 (8.0–12.9) | 3.1 (1.5–5.2) | 0.5 (-1.6–1.7) |
| ScvO$_2$ (%) N baseline = 4 N after bolus = 3 N after 1 hour = 3 | 65.0 (62.2–71.1) | Too small sample size | Too small sample size |

Values are presented as median with (IQR).

MAP = Mean arterial pressure. UO = Urine output. HR = Heart rate. CI = Cardiac index. CVP = Central venous pressure. ScvO2 = Central venous oxygen saturation. N/A = Not applicable.

one that might be administered without much consequence. If this is the case it might explain some of the FBT expectations. However, such thinking would be problematic in light of the connection between positive fluid balance and long-term morbidity and mortality [7, 11]. More research is needed to unravel such aspects of the psychology of decision-making leading to FBT prescription.

## Relationship with previous studies

Fluid responsiveness has been extensively studied and is a well-accepted concept. However, the number of patients that are fluid responders in the ICU remains low. Also, measurements of CO and SV are often not available in patients who are considered in need of FBT. In this study, we have therefore focused on the goals that intensivists themselves have chosen as target, an assessment that better describe if clinically relevant goals are met, as it takes intensivists' evaluation and judgement into account.

To our knowledge, no previous studies have investigated the accuracy of expected treatment response to FBT. Unfortunately, while the proportion of fluid responders has been

described as 50% in the ICU, the accuracy of predicting this response after FBT was only in the range of 8–52% depending on the parameter chosen. This is disappointing but aligned with other areas of clinical practice. For example, palliative care doctors predicted patient survival inaccurately in 34% of cases and overestimate survival time in 51% of cases [13].

As in the FENICE trial, MAP was the commonest trigger for FBT [9] in our cohort, and the use of more advanced parameters for guiding FBT remained low. This shows the continuing problem of using inaccurate parameters to predict and measure response to fluid administration. However, in the FENICE study, CVP was commonly used [9], while in our population, CVP was used in only 3% of cases. This likely reflects changes in practice due to evidence against using CVP as a marker for fluid administration [3, 6, 14]. Also, in our study, compared to both FENICE and international surveys, bolus volume was both slightly smaller and administered over longer period [8, 9].

## Study strengths and limitations

The strengths of our study include the pragmatic and novel study design. Another strength of our study is that we corrected statistically for changes in vasopressors and sedatives to exclude them as potential confounders. Also, our study recruited from two ICUs in two different countries, both a university centre and a tertiary centre. This reduces the risk that local treatment algorithms and traditions influenced the results and adds external validity. Moreover, we evaluated all effects for up to one hour after FB, as previous trials have shown a decrease in the effect of fluid in this timeframe [15, 16]. Our study also included evaluation of UO, which is one of the most commonly used triggers for guiding FBT and an important clinical marker for renal function.

A limitation of our study is that the sample is a convenience sample and relatively small. Also, we had originally planned for a sample of 100 patients, but due to issues with the equipment used to extract the monitoring data we unfortunately had to end the study at 77 patients. However, the findings are clear and unlikely to be materially altered by a larger sample.

Compared to some previous trials defining a fluid bolus and the standard definition of fluid responsiveness, the mean bolus size in this group was smaller, and given rather slowly (300 ml given over 21 minutes). This might affect the absolute size of the fluid response and would perhaps mean that fewer patients would have a rise in SV or CO > 10%. However, the intensivists chose both the fluid volume and rate of infusion and had this in mind when considering their expectations. Thus, their predictive accuracy is adjusted to that bolus size and reflects clinical practice. We described accuracy both as the percentage of expectations that were within the interval of +/-5%, as well as by mean bias in standard Bland-Altman plots with wider limits of agreement. These results were slightly divergent, probably due to a larger effect of outliers on the mean bias in the Bland-Altman plots. However, both methods show that it is difficult to estimate what the effect of a FB on MAP, HR and UO will be.

## Conclusion

Hypotension was the most common indication for giving fluids, yet MAP expectations were met in only 25% of cases. The other common reasons for fluid boluses were poor urine output and tachycardia, with expectation met even less frequently. The clinical expectations of intensivists were not met to a substantial extent and showed poor accuracy at the end of the bolus and after one hour. Since the clinical effect is often small and does not meet clinical expectations, a reassessment of the rationale, effect, duration, and role of FBT in ICU patients appears justified.

## Supporting information

**S1 Checklist. STROBE statement—checklist of items that should be included in reports of observational studies.**
(DOCX)

**S1 Appendix.**
(DOCX)

**S1 Protocol.**
(DOCX)

## Acknowledgments

We wish to thank the staff at Södersjukhuset Hospital, Stockholm, Sweden and Austin Hospital, Melbourne Australia.

## Author Contributions

**Conceptualization:** Olof Wall, Anthony Wilson, Glenn Eastwood, Rinaldo Bellomo, Maria Cronhjort.

**Data curation:** Olof Wall, Salvatore Cutuli, Adam Lipka-Falck.

**Formal analysis:** Olof Wall, Maria Cronhjort.

**Funding acquisition:** Olof Wall, Maria Cronhjort.

**Investigation:** Olof Wall, Salvatore Cutuli, Anthony Wilson, Glenn Eastwood, Adam Lipka-Falck, Maria Cronhjort.

**Methodology:** Olof Wall, Glenn Eastwood, Rinaldo Bellomo, Maria Cronhjort.

**Project administration:** Olof Wall, Glenn Eastwood, Rinaldo Bellomo, Maria Cronhjort.

**Resources:** Rinaldo Bellomo, Maria Cronhjort.

**Software:** Adam Lipka-Falck.

**Supervision:** Rinaldo Bellomo, Maria Cronhjort.

**Validation:** Olof Wall.

**Writing – original draft:** Olof Wall.

**Writing – review & editing:** Olof Wall, Salvatore Cutuli, Anthony Wilson, Glenn Eastwood, Daniel Törnberg, Rinaldo Bellomo, Maria Cronhjort.

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
