## [Decision Letter · Decision Letter 0]

30 Dec 2021

PONE-D-21-33377An observational study of intensivists’ expectations and effects of fluid boluses in critically ill patientsPLOS ONE

Dear Dr. Wall,

Thank you for submitting your manuscript to PLOS ONE. After careful consideration, we feel that it has merit but does not fully meet PLOS ONE’s publication criteria as it currently stands. Therefore, we invite you to submit a revised version of the manuscript that addresses the points raised during the review process.

We look forward to receiving your revised manuscript.

Kind regards,

Jaishankar Raman, MBBS, MMed, FRACS, PhD

Academic Editor

PLOS ONE

Journal Requirements:

Additional Editor Comments:

Thanks very much for the submission. Please make the requisite minor revisions requested by each of the reviewers.

Congratulations on the acceptance of this paper.

Reviewers' comments:

Reviewer's Responses to Questions

**Comments to the Author**

1. Is the manuscript technically sound, and do the data support the conclusions?

Reviewer #1: Yes

Reviewer #2: Partly

Reviewer #3: Yes

2. Has the statistical analysis been performed appropriately and rigorously? 

Reviewer #1: Yes

Reviewer #2: Yes

Reviewer #3: Yes

3. Have the authors made all data underlying the findings in their manuscript fully available?

Reviewer #1: No

Reviewer #2: Yes

Reviewer #3: Yes

4. Is the manuscript presented in an intelligible fashion and written in standard English?

Reviewer #1: Yes

Reviewer #2: Yes

Reviewer #3: Yes

5. Review Comments to the Author

Reviewer #1: The authors conducted a prospective, observational multi-center cohort study to investigate, whether the quantitative expectations of treating intensivists coincide with the actual observed effects of FBT in critically ill patients. The authors included 77 patients.

They concluded that physiological expectations of intensivists after FBT carried a high risk of both over and underestimation, so agreement seem to be rather weak.

The data and presentation are suitable, to support the conclusion. The analysis of the data is by appropriate sound scientific descriptive methods.

I do have some minor comment, which the authors may consider.

The Bland Altman approach is suitable here, but leaves the reader with the two agreement axis accuracy and precision. Lin's concordance coefficient would summarize the two axis. There is also a version for repeated measures similar to Bland Altman. May be the authors would like to consider this measure, with 95% CI.

L29: 77 patients are enrolled, but 100 are planned in the protocol. It would be informative to discuss this difference. (See patient flow chart later)

L39: The authors define the cutoff by 5% above or below the measured values. The problem with this definition is, that there will be almost always observations outside the 5% margin defined by the empirical distribution. Although Bland Altman's approach is data driven as well the cutoff by SD seem to be more robust. Could you please clarify the definition?

L126: Some more details are necessary to describe the constitution of the "convenience sample" so that the effect of possible selection biases could be assessed.

L130: skip "in"

L132: may be "included in the analysis

(Comment: As this is rather a descriptive explorative analysis it suffices to describe the missing pattern. In principle multiple imputation techniques can be used to mitigate attrition bias here.)

L 145f: Add a figure illustrating patient flow similar to consort flow chart.

Reviewer #2: This multi centre observational study compares the expectations of intensivists after a fluid bolus with the actual observed effects in a mixed surgical and medical intensive care population

The manuscript is clearly written, and the design is novel and interesting, and the question important.

Comments;

-The main problem is that the very low expectations of the intensivists means it is difficult to draw conclusions about them being under or over estimations. A mean MAP difference of 2.6mmhHg is arguably not clinically meaningful. The expectations of change in the other measured variables are also very low. It begs the question - did the authors ask and measure the correct question? Eg other measures of perfusion such as capillary return (Hernandez. 2018. JAMA. 2019;321(7):654-664. doi:10.1001/jama.2019.0071). Plus, the 57% that were on NA were unlikely to have had a large MAP change - as this NA titration is usually used to maintain a MAP. This would more likely be manifested as a drop in NA as the authors point out. Although this is acknowledged by the authors, it still limits the validity of their results.

-A median bolus of 300mls is rather small to expect large and clinically meaningful changes in physiology.

-What was the rate of missing data?

Minor points

-Albumin - was the 4%? 20%? not clear

Reviewer #3: The authors have conducted an observational study to assess the expected response from fluid bolus therapy (FBT) in patients admitted to ICU with critical medical and surgical conditions. Trigger points for FBT included commonly used hemodynamic parameters or markers of tissue perfusion such as HR, MAP, CI, lactate levels or urine output. The endpoints were increase of MAP and other commonly used hemodynamic parameters and urine output on conclusion of FBT and at one hour of completion. The authors need to be congratulated on questioning the rationale of a common scenario that is practiced universally to manage hypotension and low urine output states.

The conclusions from the study suggested that the actual response did not correlate with the expected response to MAP and urine output. For the primary reason for bolus administration (actual response), the estimation was accurate in 22% of cases at FBT completion and 47% were overestimations. After one hour, the effect estimation for the primary reason for bolus administration was accurate in 29% of cases and 31% were overestimations. For the secondary reason (predicted response), accuracy was 20% at FBT completion, with 43% being overestimations. After one hour the estimation for the secondary reason was accurate in 22% of cases, with 31% being overestimations. The effect was assessed considering the concomitant administration of vasopressor and sedation where applicable.

The authors propose that more work needs to be done in this area to define the role of FBT in ICU patients who have traditionally been treated with volume replacement with or without using CVP as marker of volume status.

The authors also suggest that a measure such as FBT is not benign as positive fluid balance in ICU settings in critical patients might correlate with adverse outcomes.

Strengths:

The study seems to be well designed with well identified goals

The concept is novel with aim to correlate the expected response obtained to FBT versus the actual effect

The statistical analysis seems to outline well the findings of the study

Limitations of the study:

Observational study in small group of patients

Does not identify the response status to patients’ cardiac and renal functions

Is there any difference in the response expected between colloids and crystalloids?

Identification of difference between medical and surgical patients, as they would represent two different cohorts who might react differently to volume administration esp. in the setting of third space fluid losses (this could potentially happen in medical patients)

However, despite all these reasons, the authors have presented a simple argument to define the role of FBT in critically ill patients in ICU

My impression is that fluid responsiveness in critically ill patients is dependent on several factors that include patients’ cardiac/ renal status/ capillary leakiness etc. and future studies will need to direct their attention at comparable group of patients. I think the manuscript will improve in its value if some of these points are highlighted.

6. PLOS authors have the option to publish the peer review history of their article (what does this mean?). If published, this will include your full peer review and any attached files.

Reviewer #1: No

Reviewer #2: No

Reviewer #3: **Yes: **Pankaj Saxena

---

## [Author Response · Author response to Decision Letter 0]

31 Jan 2022

Answers to reviewers' comments for authors regarding the manuscript “An observational study of intensivists’ expectations and effects of fluid boluses in critically ill patients”

Dear Editor,

We thank you and the referees for the comments regarding our paper with the above reference number and we are very grateful that you give us the possibility to revise and improve the manuscript. We have closely looked at the reviewer’s comments and will provide answers point-by-point.

We have conformed to the style requirements to the best of our ability, and would appreciate further instructions if we are still not fulfilling them.

The datasets from this study are not completely anonymized and cannot be shared openly, per the ethical approval obtained at the Stockholm and Melbourne ethical boards. We understand this is not ideal but the after consultation with the legal staff of both the hospital and the university we are not allowed to share a “minimal” dataset either as this would still be considered personal data since it would contain data such as age, gender, date of admission to the hospital and ICU as well as type of surgery, which would be sensitive information.

Requests for access to the data should be made to the Research Data Office at Karolinska Institutet via rdo@ki.se and if permitted by law and ethical approval, decided on a case by case basis, the data can shared.

Please see above.

Please see above, restrictions apply

The ethics statement has been moved from Declarations to Methods.

Captions have been added for the Supporting Information. 

We have reviewed our list of references and not found any instances of papers that have been retracted and no replacements have been made.

5. Review Comments to the Author

Reviewer #1: The authors conducted a prospective, observational multi-center cohort study to investigate, whether the quantitative expectations of treating intensivists coincide with the actual observed effects of FBT in critically ill patients. The authors included 77 patients.

They concluded that physiological expectations of intensivists after FBT carried a high risk of both over and underestimation, so agreement seem to be rather weak.

The data and presentation are suitable, to support the conclusion. The analysis of the data is by appropriate sound scientific descriptive methods.

I do have some minor comment, which the authors may consider.

The Bland Altman approach is suitable here, but leaves the reader with the two agreement axis accuracy and precision. Lin's concordance coefficient would summarize the two axis. There is also a version for repeated measures similar to Bland Altman. May be the authors would like to consider this measure, with 95% CI. 

While the Bland-Altman model is not perfect it is useful for this type of data, and while presenting both accuracy and precision is somewhat more complex also gives the reader a fuller understanding of the findings. We collaborated with a statistician regarding the statistical models and were advised against using a correlation model as the degree of correlation would not only reflect the accuracy but also the heterogeneity and spread of our sample population. Also, to our knowledge the Bland-Altman model is frequently used for presenting accuracy and precision and might be familiar to interpret for the reader, whereas Lin´s concordance coefficient might be less so. 

L29: 77 patients are enrolled, but 100 are planned in the protocol. It would be informative to discuss this difference. (See patient flow chart later)

We agree and have added this to the Methods section (page 6 “We studied a convenience sample, planned to consist of 100 patients but resulting in 77 patients”) and Discussion in the Study strengths and limitations section (page 17, “Also, we had originally planned for a sample of 100 patients, but due to issues with the equipment used to extract the monitoring data we unfortunately had to end the study at 77 patients”).

L39: The authors define the cutoff by 5% above or below the measured values. The problem with this definition is, that there will be almost always observations outside the 5% margin defined by the empirical distribution. Although Bland Altman's approach is data driven as well the cutoff by SD seem to be more robust. Could you please clarify the definition?

We agree that defining this cutoff is difficult and problematic. The +-5% we arrived at was what we judged to be clinically relevant, even if this of course is arbitrary. Our consideration regarding the distribution was that the cutoff had to be kept narrow since the values for the parameters are highly correlated. Ie, the blood pressure of an individual patient after fluid bolus is highly correlated to the blood pressure before fluid bolus and as is seen in our material changes are likely to be slight. Therefore, using a larger cutoff might instead make it very likely for any reasonable estimation at all to be correct, which also would taint the results

L126: Some more details are necessary to describe the constitution of the "convenience sample" so that the effect of possible selection biases could be assessed. 

A description of the convenience sample has been added to Methods, please see above. Further details on patient selection are also found in the Flow Chart Fig.1 

L130: skip "in" 

Fixed

L132: may be "included in the analysis 

Fixed

(Comment: As this is rather a descriptive explorative analysis it suffices to describe the missing pattern. In principle multiple imputation techniques can be used to mitigate attrition bias here.) 

Excluded patients are presented in Fig.1, as for missing data our set was largely complete (Two datapoints missing for MAP at bolus end and after one hour for instance), and we opted against using imputation for the few missing values that we had. 

L 145f: Add a figure illustrating patient flow similar to consort flow chart.

We´re uncertain if there has been some omission of Figures, but Fig.1 should represent a Consort patient flow chart

Reviewer #2: This multi centre observational study compares the expectations of intensivists after a fluid bolus with the actual observed effects in a mixed surgical and medical intensive care population

The manuscript is clearly written, and the design is novel and interesting, and the question important.

Comments;

-The main problem is that the very low expectations of the intensivists means it is difficult to draw conclusions about them being under or over estimations. A mean MAP difference of 2.6mmhHg is arguably not clinically meaningful. The expectations of change in the other measured variables are also very low. It begs the question - did the authors ask and measure the correct question? Eg other measures of perfusion such as capillary return (Hernandez. 2018. JAMA. 2019;321(7):654-664. doi:10.1001/jama.2019.0071). Plus, the 57% that were on NA were unlikely to have had a large MAP change - as this NA titration is usually used to maintain a MAP. This would more likely be manifested as a drop in NA as the authors point out. Although this is acknowledged by the authors, it still limits the validity of their results. 

We agree that the low expectations and effects of the bolus are surprising, however to our mind these are some of the main findings of the study, and not something we created by our study design. To our knowledge this has not been described in this fashion before and is in itself worth further study. Regarding the study question, we used a questionnaire with the parameters most commonly used by the clinicians at our institutions, and the indications were then chosen by the clinicans themselves. So, if they were not correct questions and measurements then they are not used in clinical practice in either of our institutions. Specifically, regarding capillary return, this measure was not used by the physicians in either setting, however it has place when no measurement of perfusion or output is used and could have been added to the questionnaire.

The use of pressors could indeed influence the change in blood pressure, and disguise any signal in the data as NA would be lowered to keep MAP in place. We collected changes in NA as a confounding variable. We have clarified this (page 13 “Median (IQR) change in NA was 0.00 (0.00 to 0.00) after the bolus and 0.00 (-0.01 to +0.01) one hour after the bolus, which would otherwise have been a confounder for evaluating response regarding MAP” and specifically for hypotensive patients at line 256 in Results levels of NA were not lower after FB than before. 

-A median bolus of 300mls is rather small to expect large and clinically meaningful changes in physiology. 

We agree that this may be true but this is in our opinion one of the major findings of the study. Both the size of the bolus and the expectations are chosen by the clinicians studied and not the trial investigators. We might find them small and be unsurprised by the lack of response, but these were the boluses actually used in these ICU:s and the expectations the physicians had for their response.

-What was the rate of missing data?

The missing patients are presented in Fig.1. In addition to this we only included 77 rather than 100 patients due to issues with availability of the programs to extract the data from the monitors which we have expanded on in the discussion. The rate of missing data for the individual variables was low but difficult to present as it depends on the type of variable. For instance, for urine output, there were hourly data points and a few were missing. For blood pressure, the dataset is very large as it was collected per 20-second to minute and then aggregated for the analysis, so there is very little data missing (Two datapoints for MAP at bolus end and after one hour for instance) except when there was a technical issue. Also, we chose not to consider the data as missing in the cases where clinicians did not chose to use a parameter such as for instance CO or ScvO2, as this was then by their own active choice.

Minor points

-Albumin - was the 4%? 20%? not clear

Albumin was either 40 or 50g/L, this is stated in the beginning of Results and in the relevant tables

Reviewer #3: The authors have conducted an observational study to assess the expected response from fluid bolus therapy (FBT) in patients admitted to ICU with critical medical and surgical conditions. Trigger points for FBT included commonly used hemodynamic parameters or markers of tissue perfusion such as HR, MAP, CI, lactate levels or urine output. The endpoints were increase of MAP and other commonly used hemodynamic parameters and urine output on conclusion of FBT and at one hour of completion. The authors need to be congratulated on questioning the rationale of a common scenario that is practiced universally to manage hypotension and low urine output states.

The conclusions from the study suggested that the actual response did not correlate with the expected response to MAP and urine output. For the primary reason for bolus administration (actual response), the estimation was accurate in 22% of cases at FBT completion and 47% were overestimations. After one hour, the effect estimation for the primary reason for bolus administration was accurate in 29% of cases and 31% were overestimations. For the secondary reason (predicted response), accuracy was 20% at FBT completion, with 43% being overestimations. After one hour the estimation for the secondary reason was accurate in 22% of cases, with 31% being overestimations. The effect was assessed considering the concomitant administration of vasopressor and sedation where applicable.

The authors propose that more work needs to be done in this area to define the role of FBT in ICU patients who have traditionally been treated with volume replacement with or without using CVP as marker of volume status.

The authors also suggest that a measure such as FBT is not benign as positive fluid balance in ICU settings in critical patients might correlate with adverse outcomes.

Strengths:

The study seems to be well designed with well identified goals

The concept is novel with aim to correlate the expected response obtained to FBT versus the actual effect

The statistical analysis seems to outline well the findings of the study

Limitations of the study:

Observational study in small group of patients 

This is true and limits conclusions that can be drawn. Regarding the observational format though, as a main focus of our study is the clinicians expectations and goals, finding an ideal format would be difficult as this cannot be randomized or controlled

Does not identify the response status to patients’ cardiac and renal functions 

We agree that cardiac and renal function are relevant to evaluate the expected response to fluid boluses. However, we chose to focus on the parameters commonly used to evaluate the response to fluid boluses. This included diuresis, which is commonly used as a marker for renal function and in this time-span of an hour to a couple of hours is to our knowledge the most commonly used marker for renal function. Regarding the cardiac function, several measurements relating to cardiac function and perfusion such as CO, ScvO2 and lactate were used. However, we did not include echocardiographic measurements of cardiac function. This could be a weakness, but these measurements would at best be bedside echocardiographies by the physician at the bedside. They would be of very varying quality and not performed in a standardized fashion and timeframe due to the observational nature of the study, making any analysis unfeasible. 

Is there any difference in the response expected between colloids and crystalloids?

Considering the dataset consists of 77 patients already divided into sub-categories by reasons for fluid bolus administration, we chose not to further divide the groups to not make the groups any smaller than necessary and further weaken any possibility to draw conclusions. 

Identification of difference between medical and surgical patients, as they would represent two different cohorts who might react differently to volume administration esp. in the setting of third space fluid losses (this could potentially happen in medical patients). 

We agree that this is an interesting inquiry and that we would have liked to be able to compare the intensivists’ expectations in the two groups. However, our sample size is too small, especially of medical patients to allow any conclusions to be drawn from analysis. 

However, despite all these reasons, the authors have presented a simple argument to define the role of FBT in critically ill patients in ICU My impression is that fluid responsiveness in critically ill patients is dependent on several factors that include patients’ cardiac/ renal status/ capillary leakiness etc. and future studies will need to direct their attention at comparable group of patients. I think the manuscript will improve in its value if some of these points are highlighted. 

We wholeheartedly agree that these are all important factors for the fluid responsiveness of patients in the ICU setting, and that further studies are needed in the field. This trial however is more focused on the relationship between the physicians stated expectations and whether or not these goals are achieved, rather than if the patient in question is a fluid responder. To our knowledge this has been less extensively studied, and we feel that further study of what goes on at the bedside and what guides our decision making should be conducted as there is no consensus on what constituted optimal fluid therapy and there are clearly discrepancies between guidelines and actual practice.

---

## [Editor Report · Decision Letter 1]

8 Mar 2022

An observational study of intensivists’ expectations and effects of fluid boluses in critically ill patients

PONE-D-21-33377R1

Dear Dr. Wall,

We’re pleased to inform you that your manuscript has been judged scientifically suitable for publication and will be formally accepted for publication once it meets all outstanding technical requirements.

Kind regards,

Jaishankar Raman, MBBS, MMed, FRACS, PhD

Academic Editor

PLOS ONE
---

## [Editor Report · Acceptance letter]

16 Mar 2022

PONE-D-21-33377R1 

An observational study of intensivists’ expectations and effects of fluid boluses in critically ill patients 

Dear Dr. Wall:

I'm pleased to inform you that your manuscript has been deemed suitable for publication in PLOS ONE. Congratulations! Your manuscript is now with our production department. 

Kind regards, 

on behalf of

Dr. Jaishankar Raman 

Academic Editor

PLOS ONE